# A Photocatalytic Hydrolysis and Degradation of Toxic Dyes by Using Plasmonic Metal–Semiconductor Heterostructures: A Review

Shomaila Khanam [ID] and Sanjeeb Kumar Rout *[ID]

Department of Physics, Birla Institute of Technology, Ranchi 835215, India; shomaila27t@gmail.com
* Correspondence: skrout@bitmesra.ac.in

**Abstract:** Converting solar energy to chemical energy through a photocatalytic reaction is an efficient technique for obtaining a clean and affordable source of energy. The main problem with solar photocatalysts is the recombination of charge carriers and the large band gap of the photocatalysts. The plasmonic noble metal coupled with a semiconductor can give a unique synergetic effect and has emerged as the leading material for the photocatalytic reaction. The LSPR generation by these kinds of materials has proved to be very efficient in the photocatalytic hydrolysis of the hydrogen-rich compound, photocatalytic water splitting, and photocatalytic degradation of organic dyes. A noble metal coupled with a low bandgap semiconductor result in an ideal photocatalyst. Here, both the noble metal and semiconductor can absorb visible light. They tend to produce an electron–hole pair and prevent the recombination of the generated electron–hole pair, which ultimately reacts with the chemicals in the surrounding area, resulting in an enhanced photocatalytic reaction. The enhanced photocatalytic activity credit could be given to the shared effect of the strong SPR and the effective separation of photogenerated electrons and holes supported by noble metal particles. The study of plasmonic metal nanoparticles onto semiconductors has recently accelerated. It has emerged as a favourable technique to master the constraint of traditional photocatalysts and stimulate photocatalytic activity. This review work focuses on three main objectives: providing a brief explanation of plasmonic dynamics, understanding the synthesis procedure and examining the main features of the plasmonic metal nanostructure that dominate its photocatalytic activity, comparing the reported literature of some plasmonic photocatalysts on the hydrolysis of ammonia borane and dye water treatment, providing a detailed description of the four primary operations of the plasmonic energy transfer, and the study of prospects and future of plasmonic nanostructures.

**Keywords:** metal–semiconductor heterostructure; plasmonic; LSPR effect; photocatalytic hydrolysis; photocatalytic degradation; ammonia borane

## 1. Introduction

Hydrogen is a globally accepted green fuel because of its high energy density, the zero-emission capacity of toxic effluents, and it being an alternative to petrochemical resources [1,2]. Society is seeking low environmental impact fuel, and hydrogen has proven to be a crucial fuel in recent years. Unfortunately, hydrogen's safe and economical storage still creates a barrier in the path of hydrogen emerging as a viable source of energy [3]. Moreover, hydrogen applications outside of vehicular use are equally worthy of consideration; it is also a source for refining petroleum, treating metals, producing fertilizer, and processing food [4]. There are a large number of solid-state hydrogen storage materials, such as ammonia borane ($NH_3BH_3$, AB), sodium borohydride, ammonium borohydride, decaborane diammoniate of diborane, etc., that supply access for storing high gravimetric and volumetric densities of hydrogen [5]. Among all, ammonia borane is considered to be a distinguished candidate for hydrogen applications owing to its combined advantages of high hydrogen content (19.6% wt), low molecular weight (30.7 g/mol), perfect solubility

in water, relative stability in aqueous solutions concerning self-hydrolysis, and the ability to produce hydrogen at room temperature from suitable photocatalysts by a hydrolysis reaction [6,7]. Several homogeneous catalysts have been synthesized to produce three equivalents of hydrogen from one mole of ammonia borane by photocatalytic hydrolysis at an optimum temperature. For example, Goldberg and coworkers reported the fast release of hydrogen by an iridium catalyst from AB within 20 min at room temperature [8]. Baker and coworkers demonstrated that the acid-initiated dehydrogenation of ammonia borane [9] and nickel-containing catalyst are effective in the releasing 94% hydrogen within three hours by photocatalytic hydrolysis of AB [10]. Several works of literature reported by Xu and Chandra [11], Manners and coworkers [12], and Jagirdar and coworkers [13] have demonstrated the heterogeneous catalysts containing noble metals and nano ceramics for the photocatalytic activity.

In this regard, several metal–semiconductor photocatalysts have drawn global attention for wastewater treatment and solar-to-chemical energy conversion. Metal–semiconductor composites are significant candidates for the photocatalysts. A study reported by Hongwei Tian showed the electron transfer pathway of the ternary $TiO_2/RGO/Ag$ nanocomposite with enhanced photocatalytic activity and reported 91% MB degradation under visible light in 130 min [14]. Shuya Xu reported that $Ag/Ba-TiO_3$ showed an excellent photocatalytic property and degraded 83% RhB in 75 min, which is 20% more than the sole photocatalyst $TiO_3$ [15]. Similarly, Kaja Spilarewicz modified the surface of $TiO_2$ with Ag and graphene nanostructures and reported the enhanced photocatalytic performance by degrading 93% RhB in 300 min [16]. Yutong Liu and his team reported the excellent photocatalytic performance of Ag-modified ZnO and recorded 100% degradation of MB in 40 min [17].

The heterogeneous catalyst made of noble metal and semiconductors is termed as a plasmonic photocatalyst. They develop an oscillation of electrons when there is an incidence by the light wave of plasmonic resonance, which ultimately produces LSPR (a bound electromagnetic field). These materials exhibit negative real permittivity, and the best examples are silver and gold [18,19]. However, new plasmonic materials such as metal oxide, nitrides, graphene, highly doped semiconductors, and noble metal–semiconductor composites have recently been studied to show plasmonic characteristics. Plasmonic nanoceramics have proved their application from sensors to organ imaging [20,21]. The unique features of plasmonic materials to develop an enhanced electric field at the metal–semiconductor interface make it a perfect photocatalyst. Its feature of the development of photoacoustic waves and heat makes it usable in the medical field, where it is used in organ imaging and the treatment of cancer [20–22]. Metal–semiconductor composites are widely being used as photocatalysts. Photocatalysts alter the rate of photochemical reactions and can initiate the reduction and oxidation process when hit by light. It can absorb many harmful dyes and act as a disinfectant, thus helping in cleaning water [23,24]. It helps in the production of hydrogen by splitting water by initiating the reaction of dehydrogenation and liberating hydrogen from the hydrogen-stored compounds [25,26]. It can afford us with green and clean energy and sustainable treatment of water and air.

Here, in this review work, we have given a detailed description of the role of plasmonic photocatalysts (heterogeneous combination of noble metals and nanoceramics) in the efficient liberation of hydrogen from $NH_3BH_3$, AB, and photocatalytic degradation of organic dyes. Metal-based catalysts have the capability to modulate the rate of reaction and trigger solar-to-chemical energy conversion. Noble metals such as Pt, Ag, and Au and nanoparticles along with the semiconductor give a high photocatalytic performance [27,28]. However, considering the cost efficiency, the cheaper metals such as Pd, Ag, Cu, and Ni with nanoceramics can prove helpful in enhancing the photocatalytic activity. Plasmonic metal semiconductors have adjusted their role in several fields such as bio-imaging, sensing, energy storage, and photocatalysis [29]. There are several recent reports on metal-based heterostructure semiconductors for hydrogen evolution and dye water purification [30,31]. Herein, recent advancements made in the field of photocatalysis

using metal–semiconductor hybrids are discussed, with emphasis on the significant features of plasmonic metal–semiconductor nanostructures (e.g., size, shape, and composition) that influence the optical properties of the photocatalysts. The four essential mechanisms of catalytic reaction due to the plasmonic metal–semiconductor photocatalyst are explained. Until now, no review work has described the main features of metal–semiconductors that affect their localized surface plasmon resonance (LSPR) and the catalytic reaction. Some examples of recent photocatalysts used for the hydrolysis of ammonia borane and photocatalytic destruction of RhB and MB are also summarized in this work.

## 2. Plasmonic Dynamics

The plasmonic metal–semiconductor composites generate localized surface plasmon resonance (LSPR) and a near-electric field at the junction between the semiconductor and metal (Schottky junction), allowing hot electrons to transfer from the metal to the semiconductor and vice versa [32,33]. It is a localized effect due to its dependency on their structure and size [34]. The light wave travelling towards the plasmonic nanostructures is of two kinds of modes, TE mode and TM mode, in an electromagnetic wave travelling towards the plasmonic nanocrystals. The transverse electric mode has an electric field component perpendicular to the propagation wave vector. The transverse magnetic mode has an electric field along the direction of the propagation of light. The plasmonic wave propagates in the *x*-axis and decays in the z-direction.

For the TE wave, the electric field and magnetic field equation can be written as—

$$Z > 0, \ \tilde{E} = E_1 \vec{y} e^{i(k_x x - \omega t)} e^{-a_1 z} \tag{1}$$

$$Z < 0, \ \tilde{E} = E_2 \vec{y} e^{i(k_x x - \omega t)} e^{a_2 z} \tag{2}$$

We have from Maxwell's equation:

$$\nabla \times E = -\mu_o \frac{\partial H}{\partial t} \qquad \frac{\partial}{\partial t} = i\omega, \ \nabla \times E = \mu_o i\omega \vec{H}$$

The magnetic equations are

$$Z > 0, \ H_1 = \frac{1}{\mu_o i\omega}[a_1 \hat{x} + ik_x \hat{z}]E_1 e^{i(k_x \hat{x} - \omega t)} e^{-a_1 z} \tag{3}$$

$$Z < 0, \ H_2 = \frac{1}{\mu_o i\omega}[-a_2 \hat{x} + ik_x \hat{z}]E_2 e^{i(k_x \hat{x} - \omega t)} e^{a_2 z} \tag{4}$$

At *z* = 0, H at the tangential surface should be continuous, but these equations do not satisfy the boundary condition. Therefore, TE mode is not compatible with surface plasmon resonance.

For the TM mode, the equation of magnetic field is given by:

$$Z > 0, \ \vec{H_1} = H_1 \hat{y} e^{i(k_x \vec{x} - \omega t)} e^{-a_1 \vec{z}} \tag{5}$$

$$Z < 0, \ \vec{H_2} = H_2 \hat{y} e^{i(k_x \vec{x} - \omega t)} e^{-a_2 \vec{z}} \tag{6}$$

From the Maxwell equation:

$$\nabla \times H = \varepsilon \frac{\partial E}{\partial t} \tag{7}$$

Therefore, the solution of the equation is given by:

$$Z > 0, \ E_1 = \frac{1}{\varepsilon i\omega}[-a_1 \vec{x} - ik_x \vec{z}]H_1 e^{i(k_x \vec{x} - \omega t)} e^{-a_1 \vec{z}} \tag{8}$$

$$Z < 0, \ E_2 = \frac{1}{\varepsilon i \omega} [a_2 \vec{x} - ik_x \vec{z}] H_2 \ e^{i(k_x \vec{x} - \omega t)} e^{-a_2 \vec{z}} \tag{9}$$

At $Z = 0$, $H_1 = H_2$, the boundary condition is satisfied. Therefore, TM polarised light can produce surface plasmon oscillation. For the TM wave, we also need $E_1 = E_2$.

Therefore, $\frac{a_1}{\varepsilon_1} = -\frac{a_2}{\varepsilon_2}$. Thus, $E_1$ or $E_2 < 0$. This equation shows that the surface wave exists only at the interface between materials with opposite signs of the real part of their dielectric properties. From the Maxwell equation, we have one more condition to be met:

$$a^2 - k_x^2 = \frac{\omega^2}{c^2} \varepsilon_r \mu_r$$

$$Z > 0, \ a_1^2 - k_x^2 = \frac{\omega^2}{c^2} \varepsilon_1 \mu_1 \tag{10}$$

$$Z < 0, \ a_2^2 - k_x^2 = \frac{\omega^2}{c^2} \varepsilon_2 \mu_2 \tag{11}$$

Combining and solving Equations (10) and (11), we obtain the dispersion relation of the surface plasmons to be $K_x = \frac{\omega}{c} \sqrt{\frac{\varepsilon_1 \varepsilon_2}{\varepsilon_1 + \varepsilon_2}}$. Plasma wave has a higher wave factor when propagating. The propagation constant of the wave and the surface plasmon has to be equal so that the coupling between the evanescent wave and the surface plasmon exists. The LSPR mechanism in the plasmonic nanocrystals can be best explained by Drude's model. The Drude's permittivity of metal is written as the complex quantity: chemistry-1709450-image01

$$\varepsilon(\omega) = e + i\dot{\varepsilon} \text{ and } \varepsilon(\omega) = \varepsilon_\infty - \frac{\omega_p^2}{\omega^2 + \gamma^2} + i \frac{\omega_p^2}{\omega(\omega^2 + \gamma^2)} \gamma$$

where $\omega_p$ is the plasma frequency (highest frequency which the electron could be able to respond with), and $Y$ is the damping factor. The plasma frequency is defined as $\omega_p^2 = \frac{e^2 n_e}{m^* \varepsilon_o}$, where $n_e$ is the free carrier density, m* is their effective mass, and gamma (damping factor) is defined as $\gamma = e/\mu m^*$. Here, $\mu$ is the mobility of the carrier. The permittivity of the material is its overall response to the external field. The real part of permittivity describes the refraction of light, and the imaginary part represents the losses of the materials, and it increases with decreasing frequency. The real part of the permittivity must be negative, and the imaginary part must be close to zero to support the plasmonic effect. This condition gives strong resonance and thereby strong absorption of the incident light. When the TM polarised light illuminates the plasmonic nanocrystals, the total field generated around the nanocrystals will be the combination of both the external field and the crystal response field. This will pull off the electrons from the nucleus into equilibrium. The pulled-off electrons and nucleus induce a coulomb field, which is the same but opposite in direction; this will cancel the two fields, and no field shall pass out thus, making a shield of the electric field at the metal and dielectric interface as shown in Figure 1a,b. The pioneering work of Zhao et al. in 2009 [35] showed plasmonic nanoparticles exhibit the unique property of creating an evanescent electric field when electromagnetic light of a wavelength larger than the size of the particle and frequency $\omega = \omega_p$ (where $\omega_p$ is plasma frequency) is made incident on the noble metal surface. The electrons of the metals start oscillating with resonating frequency. This oscillating electron is called a plasmon. This coupling of polaritons (electrons and photons) at the metal–dielectric interface is called surface plasmon resonance. The SPR enables a strong absorption and moderate scattering of incident light. This can be measured by observing the absorbance intensity in a UV spectrometer. The SPR band intensity and wavelength are dependent on the properties of the particle, including shape, structure, metal type, size, and dielectric material. There are two fundamental excitations, SPP (surface propagation plasmons) and LSPR (localized surface plasmons). SPP is the evanescent propagating plasmons. Under the resonant condition, a travelling

surface wave is generated along the surface. A localized surface plasmon is a surface plasmon geometrically restricted to a small object, such as a nanoparticle or point. Noble metals and semiconductor hybrids also show a promising LSPR.

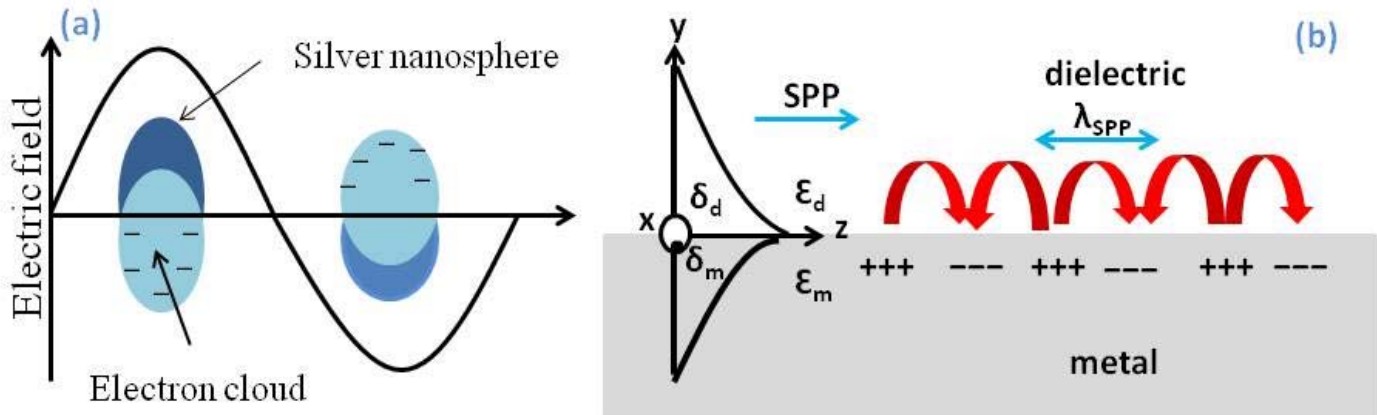

**Figure 1.** Electric field formation in a (**a**) nanosphere and (**b**) metal rod.

### 3. Synthesis of Plasmonic Metal–Semiconductor Photocatalyst

The synthesis procedure plays an important role in describing the characteristics of an excellent plasmonic photocatalyst. A promising plasmonic photocatalyst with high LSPR should produce more charge pairs and the least recombination of electrons. However, the LSPR effect of a plasmonic photocatalyst is totally influenced by the size, shape, geometry, and amount of metal–semiconductor loading. Each of these factors can be tuned to change the resonance wavelength, which depends upon its radius and its material composition; increasing the radius will increase the wavelength at which plasmon resonance occurs, thus showing the red shift in LSPR spectra. In gold nanorods, the particle shows two SPR wavelengths transverse and longitudinal, as shown in the Figure 2a,b. The longitudinal SPR of the plasmonic gold nanorod occurs at a higher wavelength, indicating that it is easily polarised. The longitudinal SPR is more sensitive to aspect ratio (ratio of length to width) than transverse SPR. The longitudinal LSPR of gold nanorods can be tuned between 500 nm and 2000 nm by adjusting to longer aspect ratios, while the transverse LSPR remains constant between 510 nm–520 nm.

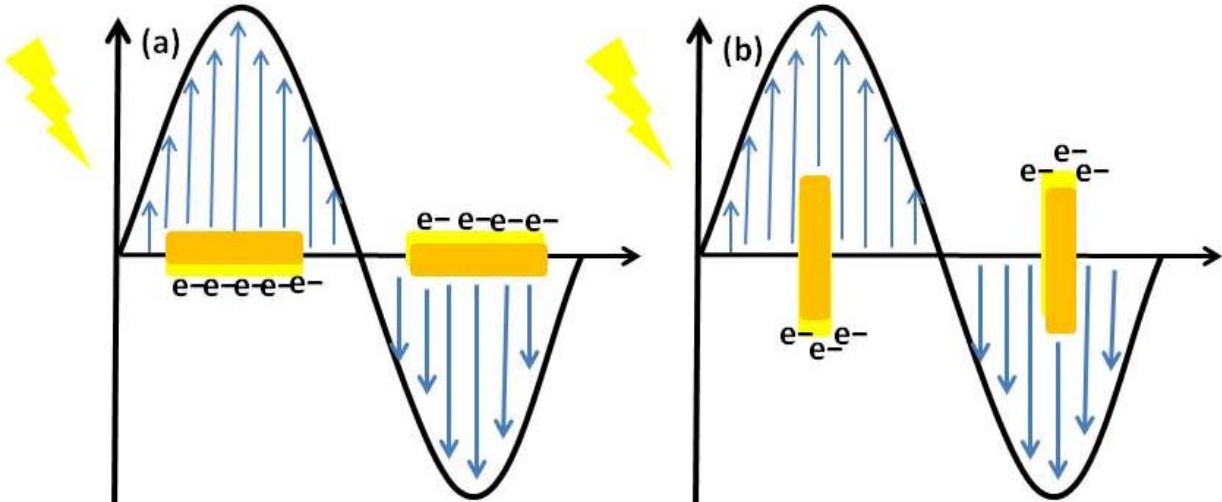

**Figure 2.** (**a**) Transverse and (**b**) longitudinal SPR of the gold nanorod.

The synthesis procedure of plasmonic photocatalysts includes two steps: the first step being the preparation of semiconductors by sol-gel, hydrothermal, solvothermal, etc., followed by incorporation of metal via chemical vapour deposition, photo deposition, and wet impregnation method. The synthesis of a $Ag/Bi_2WO_6$ plasmonic photocatalyst included hydrothermal synthesis of $Bi_2WO_6$, and Ag was photo-deposited under xenon-lamp irradiation for an hour [36]. In the literature reported by Seongwon Jo, the $Au/TiO_2$ plasmonic photocatalyst involved the chemical route for the preparation of $TiO_2$ and followed the photo deposition method to introduce Au on $TiO_2$ [37]. In another literature reported by V.I. Simagina, the $Ag/TiO_2$ photocatalyst was prepared by hydrothermal procedure, and Ag was introduced by the wetness impregnation method [38]. Haibo Yin has reported that solvothermal preparation of a hybrid of tungsten and molybdenum oxide also showed the plasmonic result [39]. Yasutaka Kuwahara has reported that the photo-deposition of Ag on Ti incorporated with mesoporous silica has resulted in a good photocatalytic behaviour [40]. Hefeng Cheng reported that the solvothermal method of preparation of $MoO_{3-x}$ nanosheets at 160 °C for 12 h also showed plasmonic behaviour and was successful in the hydrolysis of AB [41]. One-pot synthesis of Pt@MSN (platinum with mesoporous silica) reported by Maria Irum [42] shows a good catalytic effect. The plasmonic hybrid nanostructure of Au/ZnO, Ag/ZnO, $Au/TiO_2$, $Au/Fe_2O_3$, and Ag/ZnO has been reported to be synthesized by electrochemical or chemical reduction techniques. Generally, two preparation methods have been employed in this technique. One is immersing an already grown semiconductor into metal solution, followed by the addition of a reducing agent. For the second method, both the metal and semiconductor are pre-prepared and are assembled by molecular linkers [43–45]. Many sources in the literature have also reported the plasmonic activity of metal core and semiconductor shell structure. Au core/$TiO_2$ shell nanostructure has been prepared hydrothermally, where Au nanospheres are first prepared by reduction of $HAuCl_4$ with sodium citrate, and $TiO_2$ is coated on the grown Au nanosphere through hydrolysis of $TiF_4$. The morphology of $TiO_2$ is controlled by the amount of $TiF_4$ [46,47]. An even more complex plasmonic metal–semiconductor hybrid has been designed. $Fe_3O_4/Ag/SiO_2/Au$ nanostructures have been synthesized in multiple steps. Ag nanoparticles are first grown on pre-prepared $Fe_3O_4$ nanostructures followed by the coating of a $SiO_2$ layer on Ag-decorated $Fe_3O_4$ via sol-gel process. The prepared $Fe_3O_4/Ag/SiO_2$ is treated with allylamine hydrochloride to allow the adsorption of $[AuCl_4]^-$. This reduced $[AuCl_4]^-$ is reduced to form a Au shell. The plasmon coupling of Ag and Au in such hybrids gives rise to a large enhancement of Raman scattering [48–51]. There are various other reported methods for the preparation of plasmonic metal–semiconductor hybrids. Few of them are illustrated in this part, for briefness.

## 4. Factors Affecting Plasmonic Nanostructure

### 4.1. Size and Shape

The simulation of the LSPR effect of the different geometrical shapes of gallium spheres of five different radii calculated by discrete dipole approximation (DDA) was performed by Abella et al. They calculated the absorption efficiency ($Q_{abs}$), as a function of photon energy for the five spheres of radii 20 nm, 30 nm, 40 nm, 50 nm, and 60 nm. For the smallest size (R = 20 nm), the resonance peak appeared at approximately 4 eV. As the size of the Ga nanosphere is increased, the resonance peaks shift to the red and are ineffective to the polarization or incidence angle of the illumination. As the size of the nanoparticle increased, the dipolar resonance shifts to lower energy and a broad second quadrupolar resonance peak appeared at approximately 4 eV for the largest R = 60 nm particle (Figure 3a). The scattering behaviour of the Ga nanoparticles of different sizes on the sapphire substrate of dimension 1000 × 1000 × 80 nm was also investigated. Here, the electromagnetic wave interacted with both the nanoparticle and the substrate. Figure 3 shows the shift in resonance peak as the size of the Ga spheres changes. The interaction between the nanoparticles and the substrate depends on the size, shape, and the dielectric function of

the substrate and the nanoparticles. The pattern of red shift in the spectra of Figure 3b is similar to the spectra of Figure 3a. The addition of substrate only leads to an apparent decrease in the absorption efficiency. The red shift in the absorbance spectra is more apparent in the smallest nanoparticles; as the size of the nanoparticle increases, the distance between the "effective dipole" and its substrate image dipole increases, thereby weakening the nanoparticle and substrate interaction [52]. Figure 3c,d records the resonance peaks behaviour of the Ga nanoparticle ($\alpha$ = 0) through DDA analyses for a hemispherical Ga nanoparticle ($\alpha$ = 0). The LSPR of the hemisphere of radius 20 nm is red shifted compared to the isolated Ga sphere of 20 nm. This is due to the reduced electronic restoring force and increased anisotropic confinement. This red shift further increased in the case of hemisphere on the substrate (Figure 3d), because the equivalent dipole is closer to the surface making the interaction intense.

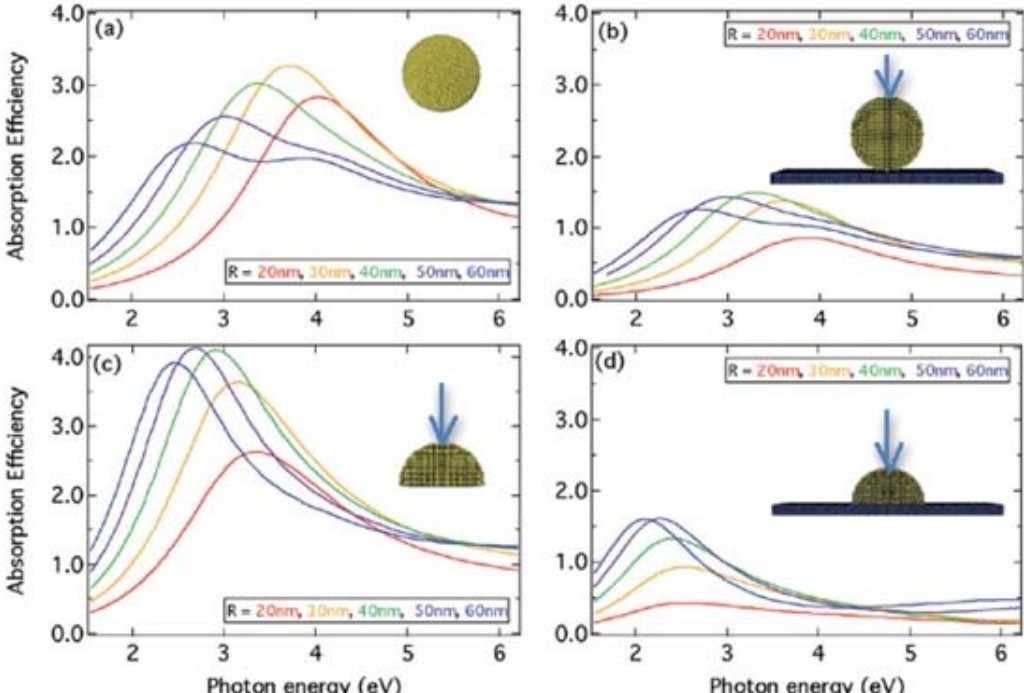

**Figure 3.** Spectral absorption efficiencies for four scattering geometries and five nanoparticle radii. The incident field is a plane wave linearly polarised at $\alpha$ = 0 (see Figure 2). (**a**) Isolated spherical Ga nanoparticle. (**b**) Spherical Ga nanoparticle located on a flat sapphire substrate. (**c**) Isolated hemispherical Ga nanoparticle. (**d**) Hemispherical Ga nanoparticle located on a flat sapphire substrate. Particle sizes are indicated in each figure: red (R = 20 nm), orange (R = 30 nm), green (R = 40 nm), violet (R = 50 nm), and blue (R = 60 nm). Figures reproduced with permission from @ACS.

Another literature source by Hui Zhang et al. also depicted a similar result when the theoretical calculation was performed on the gold of three different shapes: cube, sphere, and slab [53]. The volume of the three nanocrystals was kept the same at 5 nm³. For the slab, the width was taken to be 5 nm, and the effective area was supposed to be 5 nm. The absorption spectra depicted the strong resonance peak for the cube and sphere. In the resonance peak for the cube, the red shifts due to the increment in charge separation arising because of the sharpness of the cubic structure. The resonance peak for the gold nanosphere was observed at 525 nm, and for the gold nanocube it was observed at 575 nm (Figure 4). The confined nanocrystals showed strong LSPR compared to the slab. The LSPR effect is dependent on the size quantification factor; therefore, the slab does not offer much plasmonic resonance. In the sphere and cubic confined nanocrystals, the effects of confinement are much more robust, thus producing strong oscillations of the resonance spectra of photogenerated electrons. Thus, morphology of the composite

plays an important role in defining the plasmonic characteristic of the materials. Smaller nanoparticles exhibit a higher LSPR effect. Similarly, structures with a larger number of edges also exhibit a higher LSPR effect.

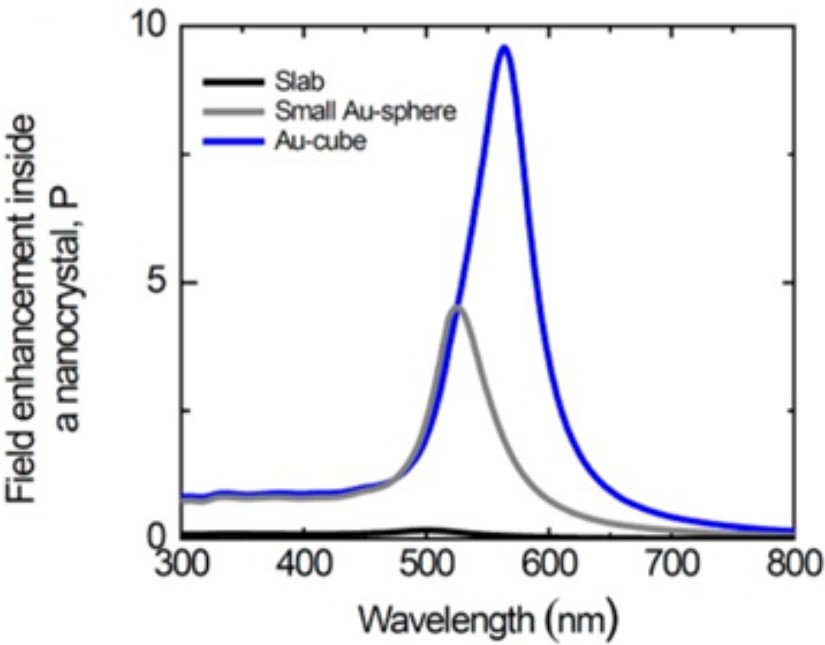

**Figure 4.** Plasmonic field-enhancement factors for the three types of nanocrystals. Figure reproduced with permission from @ACS.

### 4.2. Material Composition

The composition of the plasmonic nanostructures also plays a vital role in determining the strength and intensity of the LSPR peak. The photocatalytic splitting of water generally involves the coupling of a noble metal with a semiconductor. Now, the type of semiconductor and the type of noble metal play an important part in determining the separation of charge carriers, which will ultimately influence the amount of hydrogen and oxygen produced. Xinbo Wei et al. reported that nanostructures of $g$-$C_3N_4$ (nanospheres), $g$-$C_3N_4$/$TiO_2$ (nanofibres), $Ag$/$g$-$C_3N_4$/$TiO_2$ (nanofibres), $Pt$/$g$-$C_3N_4$/$TiO_2$ (nanofibres), and $Au$/$g$-$C_3N_4$/$TiO_2$ prepared through the electrospinning reduction and oxidation process have proved to be efficient in the photocatalytic reduction of water [54]. However, the amount of hydrogen produced is different from the different nanofibres. The intensity of the LSPR peak achieved by $Au$/$g$-$C_3N_4$/$TiO_2$ is at 625 nm among $Ag$/$g$-$C_3N_4$/$TiO_2$ and $Pt$/$g$-$C_3N_4$/$TiO_2$. $Ag$/$g$-$C_3N_4$/$TiO_2$ has an LSPR peak at 410 nm. There is hardly any LSPR peak observed in the case of $Pt$/$g$-$C_3N_4$/$TiO_2$ due to the high imaginary part of the dielectric function $Au$/$g$-$C_3N_4$/$TiO_2$ NFs (~1.19 μmol/h) > $Pt$/$g$-$C_3N_4$/$TiO_2$ NFs (~0.89 μmol/h) (Figure 5). Whereas the $TiO_2$ NFs showed $H_2$ evolution with the rate of ~0.14 μmol/h, and the $H_2$ evolution rate of $gC_3N_4$ NSs was recorded to be ~0.07 μmol/h of Pt. No LSPR peak is observed in the case of $TiO_2$ and $g$-$C_3N_4$. The $H_2$ evolution order of these plasmonic nanofibres under simulated sunlight irradiation came out to be $Ag$/$g$-$C_3N_4$/$TiO_2$ NFs (~1.50 μmol/h).

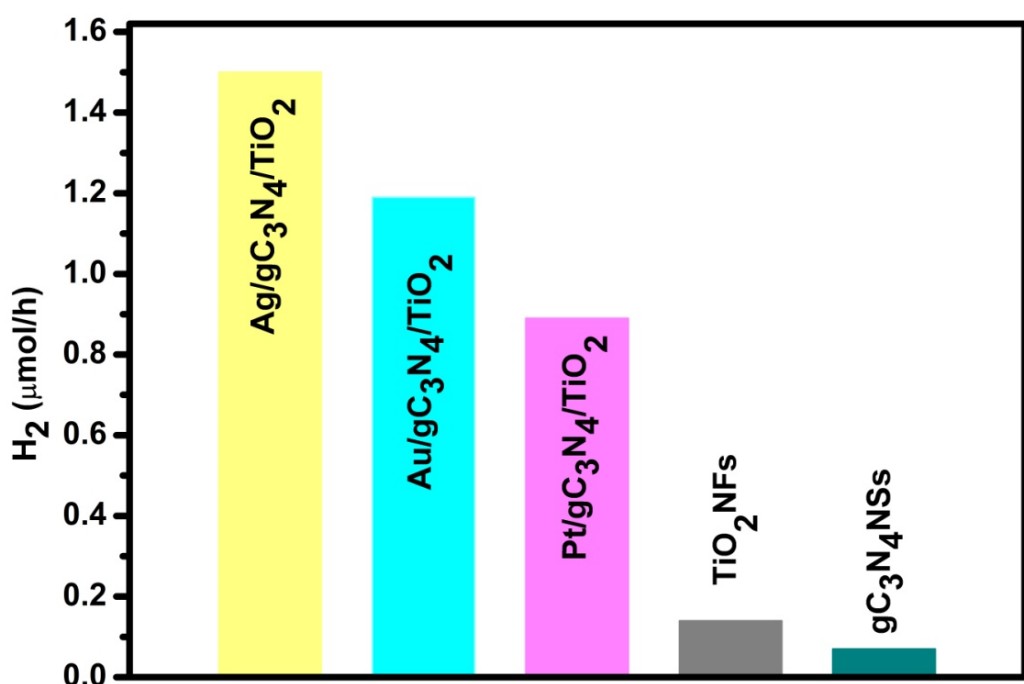

**Figure 5.** $H_2$ evolution from $NH_3BH_3$ by using $Ag/g$-$C_3N_4/TiO_2$, $Au/g$-$C_3N_4/TiO_2$, $Pt/g$-$C_3N_4/TiO_2$, $TiO_2$ NFs, and g-$C_3N_4$ NSs.

### *4.3. Ratio of Noble Metal to the Semiconductor*

The metal particle on the surface of a semiconductor proves to be an effective plasmonic nanostructure in enhancing its photocatalytic activity. The plasmonic metal can be tuned by tailoring its size, geometry, and by the ratio of the amount of it to the semiconductor. These tuning factors are important in the separation and transfer of the hole pair, the oscillation of plasmons, and the enhanced local electric field development. The difference in the tuning factor is depicted in LSPR spectra. One such example is mentioned in the literature by Shelton J. P. Varapragasam et al. $Ag/TiO_2$ nanocrystals were prepared by the technique of oleylamine/1,2-hexadecanediol reduction of $Ag^+$ on the nanorods of $TiO_2$. The ratio of Ag to $TiO_2$ was adjusted to be 1:1 and 4:1. The UV–visible spectra of the $Ag/TiO_2$ nano composite in the ratio 1:1 and 4:1 showed one LSPR, and the LSPR peak of $Ag/TiO_2$ prepared in the ratio 4:1 showed a more intense peak than $Ag/TiO_2$ of ratio 1:1. The TEM data from $Ag/TiO_2$ HNCs (1:1 ratio) indicated that the nanoparticle diameter is $3.4 \pm 0.8$ nm, and the number density is ~3.2 Ag nanoparticles per $TiO_2$ nanorod. In contrast, the $Ag/TiO_2$ HNCs (4:1 ratio) nanoparticle diameter is $4.0 \pm 1.5$ nm and has a number density of ~1.9 Ag nanoparticles per $TiO_2$ nanorod [55]. The $H_2$ evolution rate of $Ag/TiO_2$ 1:1 is 12.3 μmolg/cat/min, whereas the $H_2$ evolution rate of $Ag/TiO_2$ 4:1 is 3.3 μmolg/cat/min. This is due to the higher number density of Ag particles in the $Ag/TiO_2$ 1:1 composite. Globally, researchers are working on the perfect combination of the metal–semiconductor hybrid in order to obtain the perfect plasmonic effect, but the perfect composition for the LSPR effect is still a mystery.

### 5. Plasmonic Metal–Semiconductor Heterostructure for Photocatalytic Hydrogen Evolution from Ammonium Borane

The introduction of a plasmonic photocatalyst has evolved as an assuring technique for high photocatalytic efficiency under visible light irradiation. It is a hybrid of metal and semiconductor and can produce photoelectrons that can push many chemical reactions, where individual entities could not perform. These combined heterostructured materials produce new characteristics when placed together within close proximity. Its efficiency lies on the nature of the photosensitized catalyst, [56,57] suitable photon source

for excitation, [58] electron–hole pair generation, and [59] recombination rate of electron–hole pairs.

Hydrogen release from ammonium borane can take place by three methods, namely hydrolysis, pyrolysis, and solvolysis [12,60,61]. Among all three methods, hydrolysis is the perfect strategy for the removal of hydrogen from AB. It requires a suitable catalyst at an ambient temperature and is low energy consumption [7]. In contrast, pyrolysis involves exhaustion of energy and produces by-products that are strenuous to eliminate [9]. Furthermore, solvolysis requires an organic solvent, which also does not meet the requirement of green hydrogen [10]. The hydrolysis mechanism can generate 3 mol of hydrogen from 1 mol of AB in the presence of a catalyst. The hydrolysis equation is:

$$NH_3BH_3 + 2H_2O \rightarrow NH^{4+} + BO^{2-} + 3H_2 \tag{12}$$

The reported study showed that $NH_3BH_3$ could undergo hydrolysis and generate $H_2$ via various mechanistic pathways. An amount of 1 mol of $NH_3BH_3$ produces 3 mol of $H_2$ in the photocatalyst supported hydrogen evolution. The first one is when a plasmonic photocatalyst is irradiated; if the metal is excited, carriers and holes are pushed to the edges of the catalyst, and an electric field is created from the semiconductor towards the metal, which creates LSPR charge pairs on metal NPs, and the following electron transfer to AB can lead to the breakdown of AB. The decomposition reaction of AB pushes electrons to the lowest unoccupied molecular orbital (LUMO) leading to N–B bond elongation [40,41]. In the second case, the heterostructure of plasmonic metal and high bandgap semiconductors on illumination of the electrons are transferred from metal to the semiconductor for inducing the photocatalytic reaction [62–66]. For materials such as noble metals coupled with low bandgap semiconductors [67–69], when analyzed, it is found that there is a probability of the electron transfer from either noble metal to semiconductor or from semiconductor to noble metal, depending on the energy band positioning at the interface of the metal–semiconductor and illumination source. The case where both the metal and semiconductor are sensitive to light is more favourable for inducing a photocatalytic reaction as it can produce a greater number of charge carriers than the other two categorizations in which either the metal or semiconductor absorbs light. The LSPR effect created reduces the recombination rate of the electron–hole pair and generates a charge separation region. The generated electron ($e^-$) and hole ($h^+$) pair plays an important part in the semiconductor's photocatalytic activity, and hydroxyl radicals (•OH), formed in the photocatalytic process with $H_2O$, have a beneficial response to the hydrogen evolution from AB. Scavenger tests performed help in determining the participating species in the reaction. The addition of electron scavengers, hole scavengers, and hydroxyl scavengers in the reaction reduces the hydrogen evolution rate if $e^-$, $h^+$, and •OH are active species in the reaction. If any of the species are not participating, its scavenger will not affect the efficiency of the reaction.

Seongwon [37] investigated the effect of Au loading on $TiO_2$ ($Au/TiO_2$) catalysts with varied gold composition (0, 0.5, 1, 2, and 3 wt%) in $H_2$ evolution from AB under visible light irradiation. The $H_2$ production was maximum with 1 wt% loading (85 μmol) in 4 h showing that the optimum content of Au loading is important for the smooth reaction, and a higher content of Au loading reduces the active sites of the reaction. A similar result was published by Xinhong Qi [70], where catalytic hydrolysis of AB for hydrogen generation was performed with $Pt@SiO_2$ and $PtNi@SiO_2$ catalysts. The $PtNi@SiO_2$ catalyst was found to be more active than $Pt@SiO_2$ with a release of 4.9 mL/min and 1.1 mL/min hydrogen. The $Ag/W_{18}O_{49}$ heterostructures film obtained by assembling Ag NRs onto $W_{18}O_{49}$ NWs surface through a solvent evaporation method combined with a photoreduction process recorded enhancement of the $H_2$ evolution rate of 0.18 μmol/min. Whereas, the single Ag NRs and single $W_{18}O_{49}$ NWs film recorded a $H_2$ evolution rate of 0.01 and 0.02 μmol/min, respectively [71]. The catalytic behaviour of $Cu_{2-x}S$ nanowires with x varied between zero and one was reported by Pei-Hsuan et al. Among all the samples with different Cu–S composition, $Cu_7S_4$ exhibited the highest hydrogen evolution rate of 25.54 mmol/min, but this catalytic activity tremendously increased decorating $Cu_7S_4$ with Pd and recorded an

enhanced hydrogen evolution rate of 157.04 mmol/min [72]. The dehydrogenation rate of the $Pd/Mo_xW_{1-x}O_{3-y}$ hybrid was reported to be 5.2 times higher than $Mo_xW_{1-x}O_{3-y}$ $NaBH_4$ (6.72 mL/min) by Yin et al. [73]. Figure 6 shows the pictorial representation of photocatalytic hydrolysis of hydrogen from ammonia borane. There is much literature reporting the evolution of hydrogen with metal–semiconductor, and a few of them are listed in Table 1.

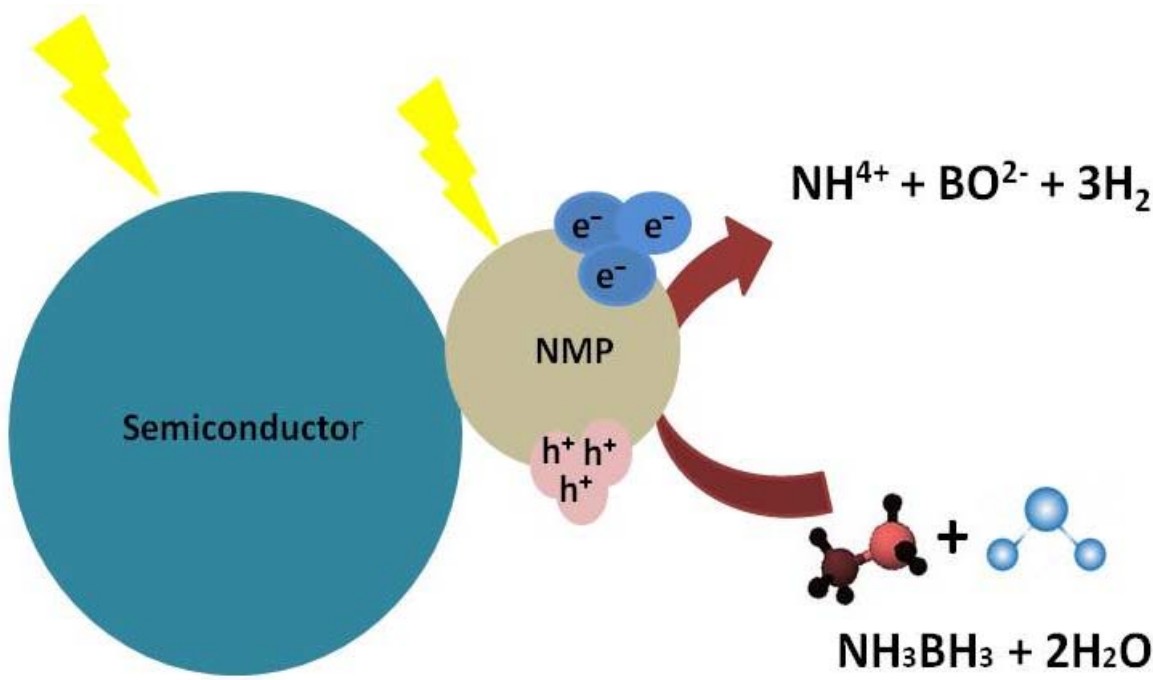

**Figure 6.** Electron–hole process in photocatalytic hydrogen evolution from $NH_3BH_3$ from metal–semiconductor photocatalysts.

**Table 1.** Efficiency of metal–semiconductor photocatalysts in hydrogen evolution from ammonia borane.

| Catalysts | Source | Time | H$_2$ (Evolution) | Ref. |
|---|---|---|---|---|
| Au/TiO$_2$ | Visible light | 4 h | 85 μmol | [18] |
| Ag/TiO$_2$ | LED (365 nm) | 3 h | 330 μmol | [38] |
| Ag/Ti/SBA | Visible light | 1 h | 65 μmol | [21] |
| Mo$_x$W$_{1-x}$O$_{3-y}$ | LED (420 nm) | 1 h | 60 mol% | [39] |
| MoO$_{3-x}$ | LED (420 nm) | 1 h | 77 mol% | [41] |
| Cu/TiO$_2$ | Without light | 1 h | 90 mol% | [7] |
| Pt-TiO$_2$ | UV light | 4 h | 15 μmol | [74] |
| ZnO-Fe$_2$O$_3$-TiO$_2$ | Visible light | 50 min | 2.7 equivalent of H$_2$ | [75] |
| PtNi/g-C$_3$N$_4$ | Visible light | 60 min | 3 mL/min | [76] |
| PtNi@SiO$_2$ | Visible light | 60 min | 4.9 mL/min | [41] |
| Pt@SiO$_2$ | Visible light | 60 min | 1.1 mL/min | [41] |
| Ag/W$_{18}$O$_{49}$ | Visible light | 60 min | 0.18 μmol/min | [71] |
| Pd/Cu$_7$S$_4$ | Visible light | 240 min | 157.04 mmol/min | [72] |
| Pd/Mo$_x$W$_{1-x}$O$_{3-y}$ | Visible light | 10 min | 6.72 mL/min | [73] |

## 6. Plasmonic Metal–Semiconductor Heterostructure for Dye Water Treatment

Plasmonic-metal-based semiconductor nanostructures have tremendously improved the photocatalytic performance of the photocatalyst and have also proved efficient in different fields such as optical, biosensing, and storage applications [77–80]. The localized surface plasmon resonance (LSPR) effect of noble-metal nanoparticles (e.g., Ag and Au) on other semiconductors has drawn the attention of many researchers to develop visible-light-driven plasmonic photocatalysts. Bare visible-light driven photocatalysts have many

drawbacks such as poor stability, poor recycling rate, and low photocatalytic efficiency in degrading azo dyes. The need for a plasmonic photocatalyst is therefore highly desirable for environmental and energy fields. The heterostructure of metal and semiconductor photocatalysis is a promising technique to overcome the energy crisis, environmental pollution, and global warming problems [49,81,82]. In 2008, Huang et al. [83] synthesized a Ag@AgCl plasmonic photocatalyst by an ion-exchange method and studied the relationship between the morphological and photocatalytic effects [84–88]. It was found that morphology affects the photocatalytic behaviour of a plasmonic photocatalyst. Similarly, the geometrical shape of the plasmonic photocatalysts affects the charge separation property. Recently, a cubic-shaped Ag@AgCl plasmonic photocatalyst was compared with the spherical Ag@AgCl of similar size. Geometrical shape with a higher surface area and active sites proved to be a better photocatalyst [89–91]. Various literature has been reported, which demonstrates that plasmonic photocatalysts show better results in the photocatalytic degradation of organic dyes. The literature by Junki Li et al. [92] studied the photocatalytic effect of $Bi_2WO_6$ and plasmonic $Ag/Bi_2WO_6$ on RhB. It was reported that bare $Bi_2WO_6$ had only 52.5% degradation efficiency in 140 min. At the same time, $Ag/Bi_2WO_6$ showed a drastic increment in the degradation efficiency and could degrade RhB up to 94%. The $Ag@Ag_2MoO_4$-AgBr composite prepared via anion exchange and photodeposition technique exhibited a superior photocatalytic reaction and degraded rhodamine B (RhB), bromophenol blue, and amino black 10b completely (100%) within 7 min [93]. Lee et al. reported 99% degradation of MB in 12 min by the $ZnO/Au15/gC_3N_430$ photocatalyst, whereas single ZnO and $gC_3N_4$ recorded 61% and 81% destruction of dye in 120 min [94]. The $Ag/CeO_2$ composite degraded RhB dye completely within 70 min and brought 97% destruction of MB dye within an interval of 60 min [95]. Figure 7 shows the pictorial representation of the photocatalytic degradation of dye molecules. Table 2 summarizes the photocatalytic degradation of azo dyes over some reported plasmonic noble metal and doped semiconductor catalysts.

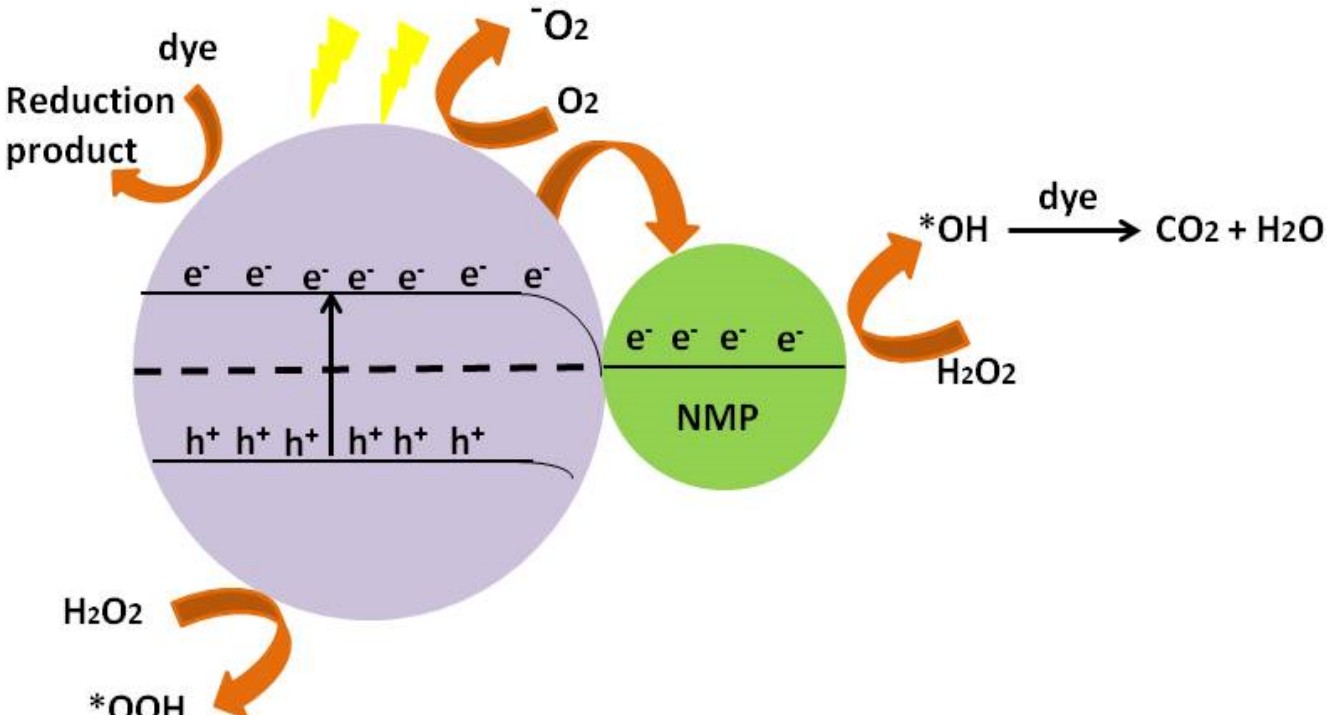

**Figure 7.** Pictorial representation of photocatalytic degradation of dye molecules under visible light irradiation in the presence of metal-semiconductor photocatalysts.

**Table 2.** Photocatalytic degradation of azo dyes over some reported plasmonic noble metal and doped semiconductor catalysts.

| Catalyst | Organic Pollutant | Time | Degradation % | Reference |
|---|---|---|---|---|
| Ag@AgCl | Phenol | 50 min | 93 | [85] |
| Ag@AgCl | DCP | 60 min | 97 | [54] |
| Ag@AgCl | AO7 | 20 min | 97 | [54] |
| Ag/AgBr-CNTs | TBP | 50 min | 100 | [96] |
| AgBr | TBP | 50 min | 60 | [62] |
| CNTs | TBP | 50 min | 45 | [62] |
| Ag/AgCl-CNTs | TBP | 50 min | 39 | [62] |
| AgCl | TBP | 50 min | 26 | [62] |
| Ag/AgI-CNTs | TBP | 50 min | 42 | [62] |
| AgI | TBP | 50 min | 8 | [62] |
| Ag-BaTiO$_3$ | RhB | 75 min | 83 | [15] |
| TiO$_3$ | RhB | 75 min | 63 | [63] |
| Ag/Bi$_2$WO$_6$ | RhB | 140 min | 94.1 | [61] |
| Au/Bi$_2$WO$_6$ | Phenol | 60 min | 93.3 | [97] |
| Bi$_2$WO$_6$ | Phenol | 60 min | 65.2 | [64] |
| Bi$_2$WO$_6$ | RhB | 15 h | 90 | [98] |
| Bi$_2$WO$_6$ | MB | 8 h | 49 | [98] |
| Ag/Bi$_2$WO$_6$ | RhB | 10 min | 50 | [99] |
| Ag/Bi$_2$WO$_6$ | MB | 90 min | 68 | [99] |
| Ag@Ag$_2$MoO$_4$-AgBr | Bromophenol blue | 7 min | 100 | [93] |
| Ag@Ag$_2$MoO$_4$-AgBr | RhB | 7 min | 100 | [93] |
| Ag@Ag$_2$MoO$_4$-AgBr | Amino black 10b | 7 min | 100 | [93] |
| ZnO/Au15/gC$_3$N$_4$30 | MB | 12 min | 99 | [94] |
| Ag/CeO$_2$ | RhB | 70 min | 100 | [95] |
| Ag/CeO$_2$ | MB | 60 min | 97 | [95] |

## 7. Proposed Mechanism

The plasmonic photocatalyst is efficient in wastewater treatment and the photocatalytic hydrolysis of hydrogen-rich compounds. There have been different mechanisms explained in the literature reported which satisfy the process. The reported literature mainly proposes four mechanisms: the mechanism followed by the plasmonic catalyst (i) Schottky barrier, (ii) increased local electric field, (iii) direct electron transfer, and (iv) plasmon resonant energy transfer.

### 7.1. Schottky Barrier

When a noble metal and a semiconductor are placed in intimate contact, a typical Schottky junction is formed. Noble metals have a high work function compared to the n-type semiconductor with an excess electron. The Fermi level of the noble metal is located below that of an n-type semiconductor. Figure 8a,b shows before and after the formation of a Schottky barrier between the noble metal and an n-type semiconductor. Upon contact between the noble metal and semiconductor, the Fermi level of the metal and semiconductor lies at the same level; electrons diffuse from the n-type semiconductor to the noble metal, creating a positively charged region in the semiconductor and a negatively charged region close to the metal. Thus, this generates a space-charge region with no free charge carriers. This leads to the formation of an electric field from the semiconductor to the noble metal. When illuminated by visible light, the internal electric field will force the photogenerated electrons from the space-charge region to move to the semiconductors and holes towards the noble metal and thus preventing the recombination of holes and electrons. The formed electron–hole pair participates in the reduction and oxidation of the chemicals. This enhances the photocatalysis efficiency of a photocatalyst. The parameter $\Phi_M$ is the work function of the metal, and $X_{SM}$ is the electron affinity of the semiconductor. Metal has lower Fermi energy compared to the semiconductor. Due to the stability, the

electrons flow from the semiconductor to the metal, and this flow will continue until the Fermi level of the semiconductor matches the Fermi level of the metal. This deformation band structure leads to the formation of a potential barrier, and electrons must overcome this barrier to flow from semiconductor to metal. This causes the accumulation of positive charge space at the interface below the semiconductor and negative charges at the metal interface. This influences the bending of the conduction band called the Schottky barrier ($\Phi_B$), explained by the following equation: ($\Phi_B$) = $\Phi_M - X_{SM}$. The Schottky barrier formed by the bending of the conduction band acts as an electron trap, stopping the electron from flowing back to the semiconductor, thus preventing the recombination of an electron–hole pair. This ultimately increases the photocatalytic efficiency of the photocatalyst.

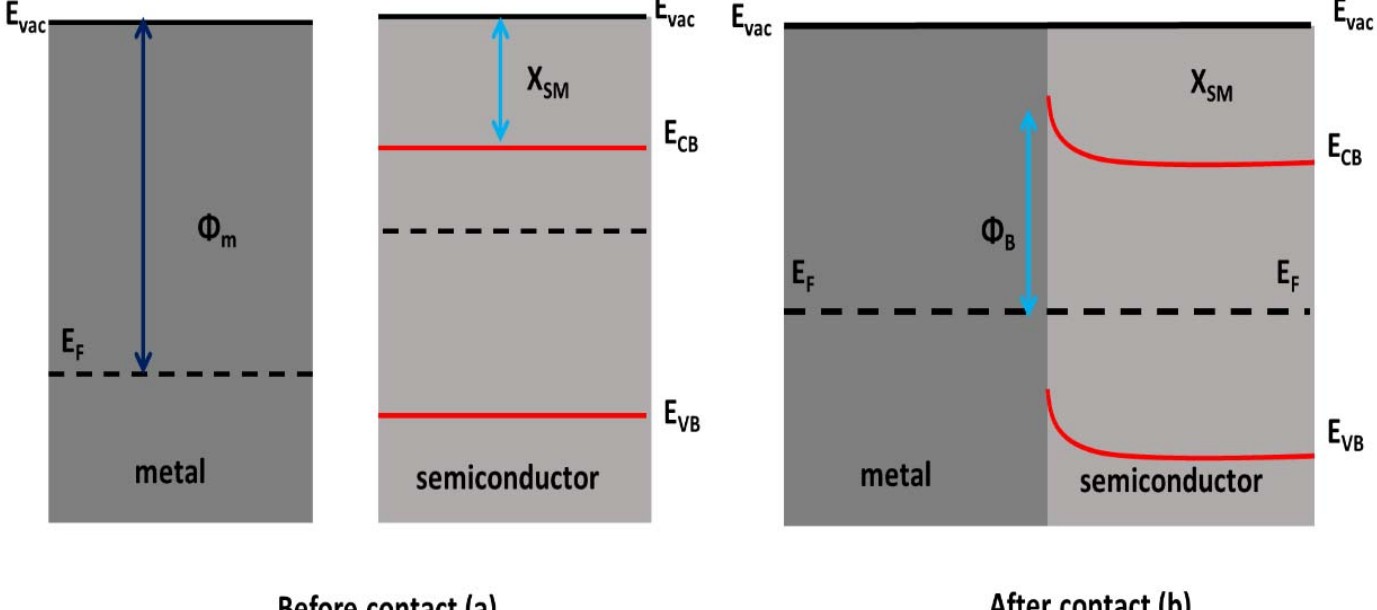

**Figure 8.** (**a**) Formation of the Schottky barrier before and (**b**) after the contact of metal and semiconductor.

## 7.2. Direct Electron Transfer

Direct electron transfer, which is also known as the LSPR sensitization effect, has been performed by Tian and Tatsuma [100,101] using a Au-TiO$_2$ composite in an electrochemical cell. In this mechanism, when a photocatalyst obtains sufficient photon energy, it excites the free electrons in the noble metal to a higher Fermi level, and hot electron holes are generated on the metal. This movement of hot electrons above the Fermi level leads to the redistribution of energy via Fermi–Dirac non-equilibrium statistics. During this redistribution, the excited electron is transferred to the semiconductor, leaving behind holes as depicted in Figure 9. The electron moves towards the semiconductor through the LSPR decay effect, as depicted in Figure 9. The metal here does not trap electrons but provides the electrons in the reaction. Tian and Tatsuma explained the effect of the metal–semiconductor in photocatalysis through this mechanism. However, some researchers disagree with this process because it is not in accordance with the Schottky barrier. Thus, there is still doubt over whether the metal acts as an electron trapper or the supplier of electrons to the semiconductor's conduction band.

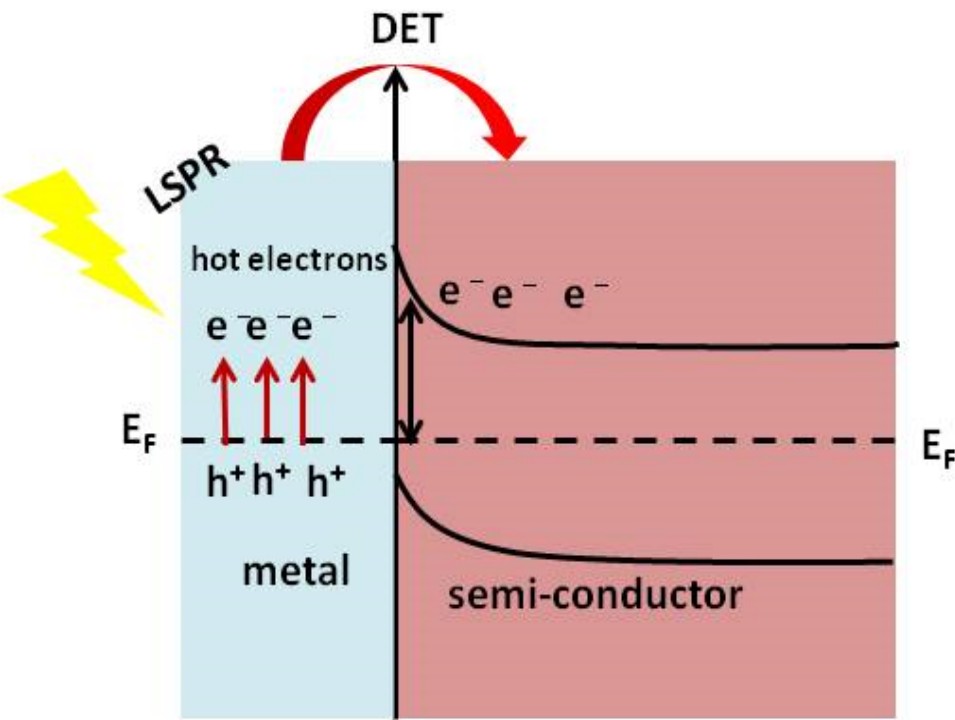

**Figure 9.** Mechanism of DET.

### 7.3. Enhanced Local Electric Field

This phenomenon has been thoroughly explained above in plasmon dynamics. When the metal–semiconductor composite is illuminated by an electromagnetic light of a frequency that equals the plasmon resonance frequency ($w_p$) of the noble metal of the composite, the incident photons are absorbed. This will lead to the formation of the oscillating electron cloud and the electric field called SPR. Now, the intensity of the electric field generated around the nanocrystal is boosted as the electric field formed is the combination of the external field and crystal response field. This enhanced local electric field leads to the increased interband transition rate making the energy generated by LSPR higher than the bandgap of a semiconductor. This enhances the photoactivity of the photocatalyst. The enhanced local electric field exists only around the noble metal, and it decays as a function of distance from the noble metals [42]. Figure 10 shows the noble metal on the semiconductor absorbs the incident light and undergoes surface plasmon oscillation, which excites electrons and holes. The space-charge region is created in the n-type semiconductor around the vicinity of the noble metal, which created an electric field pointing towards the noble metal from the semiconductor. When the electron–hole pair is excited in the space-charge region depicted as (a), the electrons are forced towards the noble metal and the holes towards the semiconductor. The semiconductor also absorbs incident light and contributes to the synergetic effect in photocatalysis.

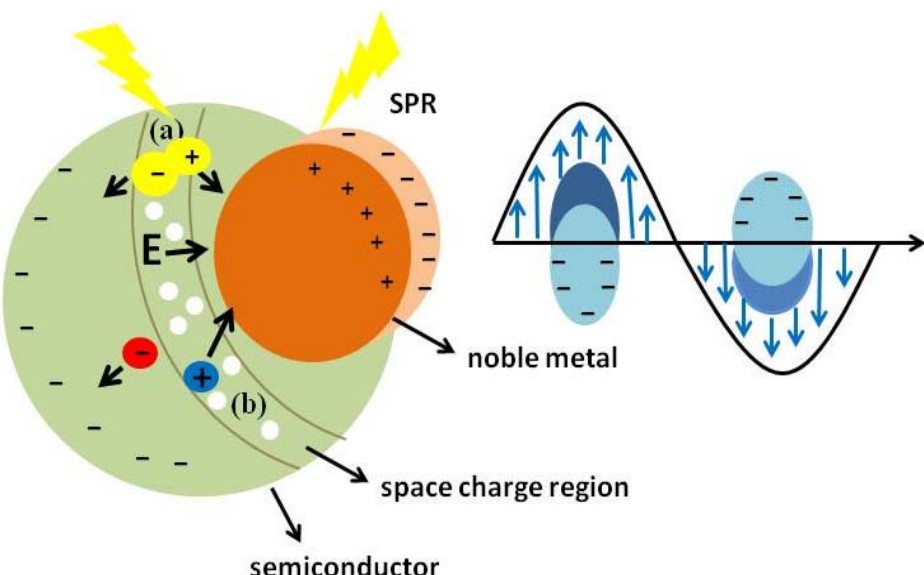

**Figure 10.** Formation of a space-charge region and pushing of electrons and holes towards the end of semiconductor and metal.

### 7.4. Plasmon Resonant Energy Transfer

When the light of the plasmon frequency falls on the metal, the electric field is generated around the vicinity of the metal. This excites the hot electron in the metal, which moves above the Fermi level of the metal and generates a hole. In the PRET mechanism, the energy is transferred from the plasmon to the semiconductor through the electric field. It refers to the dipole–dipole energy transfer. This non-radiative process leads to the coupling of the plasmonic dipole moment of the noble metal with the dipole moment of the electron–hole pair of the semiconductor. The relaxation of the localized surface plasmon dipole generates the electron–hole pair in the semiconductor. PRET is not affected by whether the semiconductor and noble metal are in contact or not. This is because the plasmon energy transfer occurs through a near-field electromagnetic interaction, as shown in Figure 11 [102].

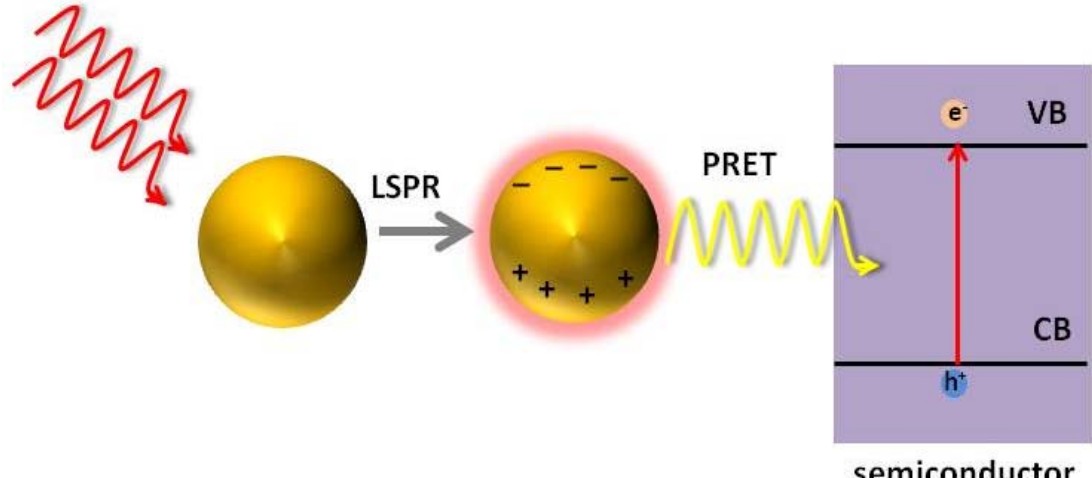

**Figure 11.** Pictorial representation of liberation of plasmon energy to the semiconductors.

### 8. Prospects and Future

The study of the mechanism of plasmonics inside a plasmonic metal–semiconductor hybrid, when light interacts with the electrons, has made it easier to develop optoelectronic devices, which can prove to be extraordinary in the future. The tuning of the size and

shape of the plasmonic material has made it advantageous in various fields where its electromagnetic field strength can be controlled. The application of plasmonics depends on the losses and cost of nanofabrication. Plasmonics can be applied from highly sensitive biosensors to invisibility cloaks. The surface plasmon resonance property of a plasmonic material can be studied thoroughly to develop nanoantennas, efficient solar cells, sensors for detecting chemical and biological agents, magnetic memory storage on disks, and other useful devices. Plasmonic switches, plasmonic nanowires, plasmonic nanocircuits, and plasmonic-assisted quantum dots are some tremendous works that are going to be launched in the near future. Plasmonics has a strong application in the medical field; plasmonic nanoparticles are tested to check their ability to treat cancer. Plasmonics has a bright future and can be widely utilized in almost every field of science.

### 8.1. Plasmonics as Energy Inputs

There has always been a growing demand for cheap and renewable energy source fuels, which derive their source from solar energy, and plasmonic hybrids can provide cheap radiation-induced energy sources. The photon–electron interaction, which takes place in a radiation-induced energy source, is strongly dependent on the optical density of states (DOS). DOS defines the number of channels that will store or allow electromagnetic energy to propagate in the medium. Plasmonic materials have attractive properties of light focusing and DOS modification, which make them perfect for application as energy inputs. The electrons' so-called plasmons oscillate when a plasmonic nanostructure is hit by an external light source, and this collective oscillation dynamic of the plasmons can be tuned by tailoring the shape, size, and area filled by the electrons. The energy stored in the DOS of surface plasmon oscillation can be converted into heat. This has paved the way to various applications such as light extraction in solid-state lighting [103,104], undercooled boiling [105–107], enhancement of frequency up-conversion processes [108], thermoelectric energy conversion [70], smart energy-saving window coatings [109,110], optical data storage [111], nanoscale heat management [112,113], and hot-electron PV cells and photodetectors [108]. Surface plasmon resonance created by the interaction of photons and electrons can be used to drive chemical reactions, which makes plasmonic nanostructures available for photocatalysis [114–116]. Plasmonic materials have the capability to concentrate light into the nanoscale, thus producing an enhanced electric field, which can cause local heating, phase transformation, and can be used to manipulate the chemical process. If the size of the plasmonic resonance is very small, then the transfer of heat is reduced [117]. This makes the localized heating of the particle possible. This localized heating creates a localized area of vapours and can even cause the melting of the substrate [118]. The conventional vapour-generating systems are not portable; they have a high cost, they work on high optical concentration, they require working fluids to be at high temperature, and storage is a big problem. These limitations of the conventional vapour system have made photothermal conversion through plasmonic nanoparticles the best alternative [119]. Although, this approach through plasmonic material is still under research. According to the recent development, plasmonics has shown excellent solar-to-heat conversion, which is a good indication that plasmonics can prove to be competitive in the solar-to-hot vapour conversion technique.

### 8.2. Plasmonic LED and Quantum Dots

QDs are artificial nanoscale semiconductor particles prepared through colloidal chemistry. The optical properties of QDs can be altered by tailoring their size and shape. The photophysics of a QD is dependent on its interaction with light. When QDs are hit by light, an electron–hole pair is created. This electron–hole pair forms a quasi-particle called an exciton. The decrease in the size of the semiconductor confines the electrons and holes, thus increasing the exciton energy. The ability of the QDs to become tuned according to the shape and size makes it eligible in multiple applications [120–122]. Plasmonic materials may revolutionize the QDs and LEDs. Figure 12 shows the plasmonic LEDs. The electric

field enhancement of plasmonic materials near the metal–dielectric interface increases the emission rate in QDs and can increase the intensity of metal–semiconductor LEDs. It has been demonstrated that the coating of LEDs with plasmonic nanoparticles increases their intensity by 14-fold. Plasmonics has made cost-effective LEDs made of silicon possible. It has been reported that covering the silicon quantum dots with silver can enhance their emission rate about 10 times. In addition, another advantage of coating with plasmonic material is that controlling the resonance frequency and the separation between the noble metal and semiconductor can increase its radiative emission by 100-fold [120–122].

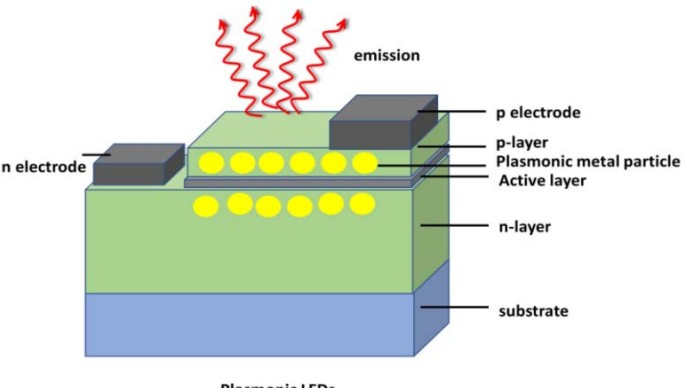

**Figure 12.** Plasmonic LEDs.

### 8.3. Communication with Plasmonics

In plasmonics, the photons excite the electron, and at a certain optical frequency, surface plasmon polaritons are set in. This SPP can be confined in an area smaller than the optical wavelength and oscillate at an optical wavelength. These features make the plasmonic materials effective in communication, as they can carry information at optical bandgap. It has been found that the appropriate design of the plasmonic metal–dielectric interface can produce a surface plasmon of the same frequency as an electromagnetic wave but with a smaller wavelength. This allows the propagation of plasmons in interconnections (nanoscale wires), which transfer information from one microprocessor to another. Figure 13 shows the traversing of SPP in the nanoantennas. In recent times, plasmonic waveguides have gained much attention as they can operate between visible and far-infrared regions. The signal processing step is performed at surface plasmon polaritons (SPP) [123,124]. The plasmonic waveguides have a transverse dimension less than 100 nm and waveguides, due to their compactness, can speed up the optical processing. Plasmonics can prove to be a boon in bridging the gap between the size mismatch of large-scale photonics and small-scale electronics.

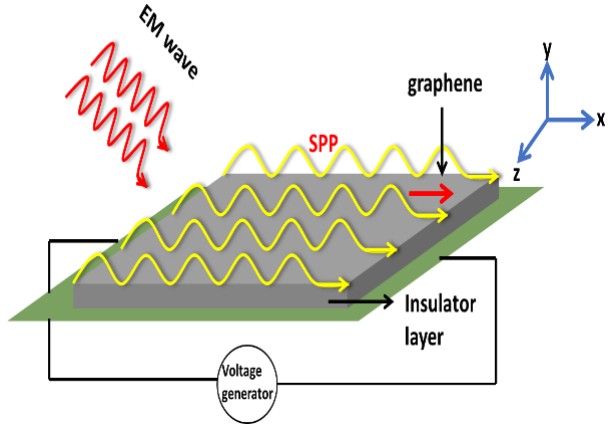

**Figure 13.** Traversing of SPP towards nanoantennas.

### 8.4. Plasmonic Nanoparticles in the Treatment of Cancer

Plasmonic nanoshells have been reported to cure cancer in mice. The plasmonic nanoshells made of gold covering silica particles of about 100 nm have been introduced in the blood of mice. These nanoshells scattered themselves around the cancerous cell of the rodents because more blood flows through the cancerous cell. When infrared light is directed towards the mice's body, the plasmonic nanoshell undergoes surface plasmonic oscillation, which ultimately will produce heat. The heat produced will raise the temperature of the cancerous cells nearby this plasmonic nanoshell. The temperature of the cancerous cell will be increased from 37 °C to 45 °C. This rise in temperature will kill the cancer cell, as shown in Figure 14, and is called photothermal killing [125–127]. Researchers are seeking permission from the authorities for a clinical trial on head and neck cancer patients. It will be a cost-effective and time-saving technique if it proves to be effective in human patients.

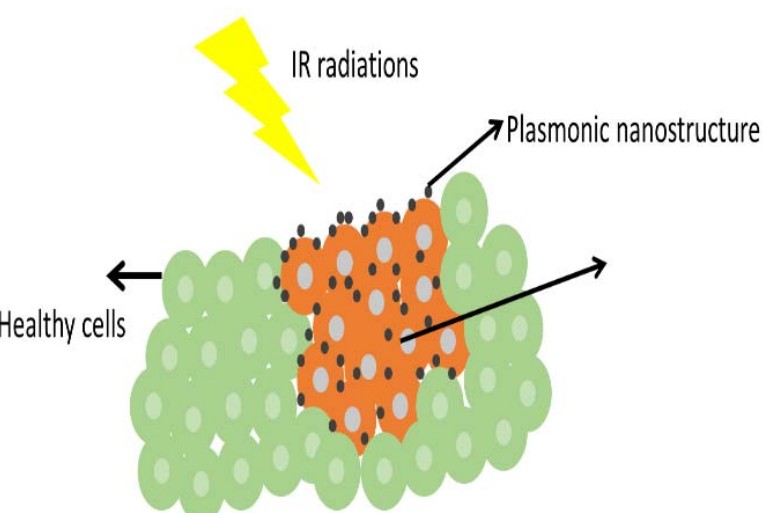

**Figure 14.** Action of plasmonic composite on cancerous cells.

### 8.5. Plasmonic in Desalination of Water

Water scarcity is a global problem, and it is also terrifying to hear that the third world war will be fought for water. Much research on desalinating water using sunlight without hampering the environment is being conducted all over the world. Plasmonics add its presence over here as well. The plasmonic solar desalination technique can increase the solar energy transfer by its unique property of enhanced light absorption and localized heating. It has been reported by Lin Zhou et al. that a plasmon-enhanced solar desalination device has been made by filling grooves in a 3D aluminium membrane with aluminium nanoparticles. Figure 15 shows the design of the device. The porous aluminium membrane filled with the aluminium nanoparticles can float on water and can absorb solar energy greater than 96% and focus the energy at the surface, which leads to the effective desalination of about 90% by localized heating of water [128]. The abundance, low cost, and healthy environment features of the plasmonic desalination device could provide us with freshwater anywhere and anytime.

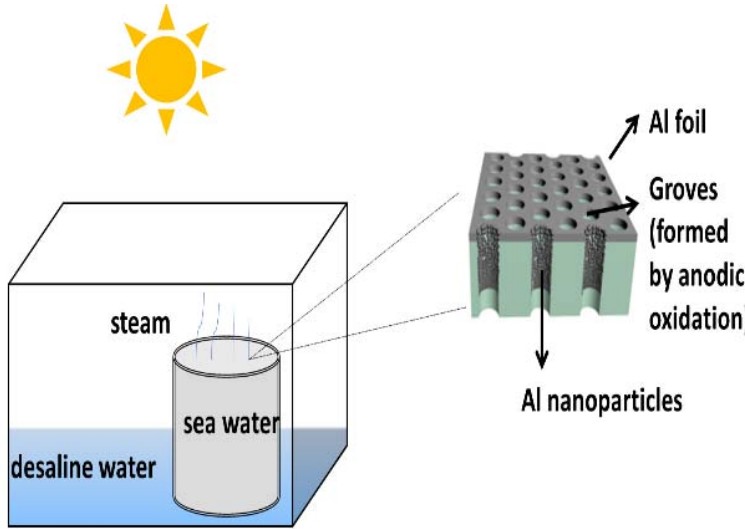

**Figure 15.** Desalination of seawater using plasmonic aluminium.

*8.6. Plasmonic Invisible Photodetector*

The plasmonic invisible photodetector, popularly known as an invisibility cloak, is an invisible machine that detects light. It is the light-detecting device that can see without being seen. It is a new division of tools that can control the flow of light at the nanoscale to perform optical and electronics functions. The device consists of silicon nanowires covered by a thin cap of gold. By tuning the metal and semiconductor, the light reflected from both the materials cancels each other, thus making the device invisible, at least to the radiations in a certain range of frequencies. The light waves in the metal and semiconductor produce an opposite dipole moment of equal strength in both the metal and semiconductor. When equally strong and opposite dipole moments meet, they cancel each other, making the system invisible. If the alignment of dipoles is not appropriate, it will lessen the invisibility [129]. Thus, having the right amount of material at the nanoscale is essential. The gold covering the silicon not only allows the light to reach the silicon where the generated current is detected, but it also makes the wire invisible. Other metals such as aluminium and copper, which are as reflective as gold, can also work in place of gold and prove to be equally effective. The cloaking effect is not dependent on the angle of incidence of light, the shape of the metal, and the placement of the metal-covered nanowires. It only depends on the tuning of the metal and semiconductors.

**9. Limitations and Challenges**

The majority of plasmonic metal–semiconductor heterostructures require noble metal cocatalysts for increasing the $H_2$ production yield and degradation efficiency, which in turn increases the cost for large-scale industrial applications. Not all the metal–semiconductor-based photocatalysts can generate $H_2$ by decomposing AB, because all the heterostructures cannot produce the LSPR effect. The LSPR intensity is totally dependent on the size, geometrical shape, and composition of the two mixtures. The right amount of noble metal is required to produce the LSPR effect at the junction of metal and semiconductors. Too much noble metal loading reduces the active site, thus hampering the efficiency of the photocatalysts. More research is to be conducted on the plasmonic photocatalyst, as a 100% result is still left to be achieved in the photocatalytic field. Plasmonic photocatalysts made of different semiconductors are modified by the size, composition, and synthesis procedure. There is still a long way to be traversed in developing the best plasmonic metal–semiconductor photocatalysts, and its application in various other fields is still to be discovered.

## 10. Conclusions

The traditional semiconductor photocatalysts suffer from poor efficiency due to the high recombination rate of electron–hole pairs and low stability. In recent times, the plasmonic metal–semiconductor heterostructure has emerged as a perfect alternative to bare semiconductor photocatalysts. The well-designed plasmonic metal–semiconductor photocatalyst can promote efficient charge separation and suppress the charge recombination rate of electron–hole pairs due to the LSPR energy transfer at the junction of the metal and semiconductor. There is a growing interest in the plasmonic metal–semiconductor nanoparticles as a catalyst in releasing hydrogen in $H_2$ from ammonia borane and in dye water treatment by converting solar to chemical energy. The perfect synthesis method of these materials is essential to fabricate a high-performance photocatalyst. However, very few such plasmonic materials have been developed to date, but their reliable photocatalytic activities motivate the designing of a wide variety of such materials.

**Author Contributions:** Conceptualization, S.K. and S.K.R.; methodology, S.K.; software, S.K.R.; validation, S.K.R.; formal analysis, S.K.R.; investigation, S.K.; resources, S.K.; data curation, S.K.; writing—original draft preparation, S.K.; writing—review and editing, S.K.; visualization, S.K.; supervision, S.K.R.; project administration, S.K.R.; funding acquisition, S.K.R. All authors have read and agreed to the published version of the manuscript.

**Funding:** This work was financially supported by the Department of Science and Technology, Govt. of India, under the WOS-A scheme (SR/WOS-A/CS-128/2018).

**Data Availability Statement:** Not applicable.

**Acknowledgments:** We would like to show our gratitude to Central Instrumentation Facility Lab of Birla Institute of Technology, Mesra for extending its support in my research work.

**Conflicts of Interest:** The authors declare that they have no known competing financial or personal interests that could have influenced the work reported in this paper.

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
