# Peer review of "A Photocatalytic Hydrolysis and Degradation of Toxic Dyes by Using Plasmonic Metal–Semiconductor Heterostructures: A Review"

_chemistry, doi:10.3390/chemistry4020034_

Round 1

Reviewer 1 Report

The submitted manuscript deals with highly timely topic of photocatalytic hydrolysis and degradation of toxic dyes by using plasmonic metal-semiconductor heterostructures. Despite to the quite some efforts of the authors, in my opinion, the article is written rather carelessly and should be corrected. Overall, I would recommend the presented manuscript by Shomaila Khanam and Sanjeeb Kumar Rout for publication in the Chemistry journal only after the following major issues are taken into account in a revised version.

Comments for authors listed below:

  1. First of all, it should be noted that there are many typos in the article:

- Starting right mistakes in author affiliation – «Birla In statute of Technology»- should be Institute of Technology I guess.

- page 2 – line 75 – «interface makes it's» - should be «interface makes it»

- in a References part it is a complete mess – all the references done in a different ways – somewhere the name of the Journals are missing, somewhere the names of the authors are missing, etc. For example reference 21 - Manuscript, A. Materials Chemistry A. 2017, doi:10.1039/C7TA07264C.

- Fig.1a Page 3 – The «silver nanosphere» and «electron cloud» are mixed up

  1. Introduction part

- First of all I don't really get what is really the aim of this article. What is a difference from the reviews already presented in the literature? Authors should rewrite the Introduction part in a way to clearly claim what is the main idea of presented review. Currently it is looks like just a description of set of the articles devoted to the metal-based heterostructure semiconductors used for photocatalysis applications.

- Paragraph 67-84 – at least a few references should be added.

- In the end (lines 90-91) there is the sentence: «In addition, the Pt-based photocatalysts can finish the reaction within a minute.». For me it is absolutly don't really have any sence – Which reaction? What are conditions? Is it really fast enough?  

  1. Part 3. Synthesis of plasmonic metal-semiconductor photocatalyst. There is just a short explanation of how the different parameters of synthesized catalysts are affecting the LSPR effect. Then the several methods of the metal-based heterostructure semiconductors just listed, even without the references. And finally just enumeration of the five works where some systems investigated is done. From my opinion this part should be expanded.
  2. In averege – from my opinion each numbered section should end with some kind of subconclusion.

Author Response

We want to thank all reviewers for their valuable comments and suggestions. We have revised the manuscript according to the reviewer’s suggestions. Text and references has been thoroughly revised in the manuscript. Other comments have also been accepted and revised in the manuscript.

Reviewer 1

  1. First of all, it should be noted that there are many typos in the article:

- Starting right mistakes in author affiliation – «Birla In statute of Technology»- should be Institute of Technology I guess.

- page 2 – line 75 – «interface makes it's» - should be «interface makes it»

- in a References part it is a complete mess – all the references done in a different ways – somewhere the name of the Journals are missing, somewhere the names of the authors are missing, etc. For example reference 21 - Manuscript, A. Materials Chemistry A. 2017, doi:10.1039/C7TA07264C.

- Fig.1a Page 3 – The «silver nanosphere» and «electron cloud» are mixed up

Reply: Authors want to thank the reviewer for the comment. All the typos have been corrected in the manuscript.

  1. Introduction part

- First of all I don't really get what is really the aim of this article. What is a difference from the reviews already presented in the literature? Authors should rewrite the Introduction part in a way to clearly claim what is the main idea of presented review. Currently it is looks like just a description of set of the articles devoted to the metal-based heterostructure semiconductors used for photocatalysis applications.

- Paragraph 67-84 – at least a few references should be added.

- In the end (lines 90-91) there is the sentence: «In addition, the Pt-based photocatalysts can finish the reaction within a minute.». For me it is absolutly don't really have any sence – Which reaction? What are conditions? Is it really fast enough?  

Reply: Authors want to thank the reviewer for the comment. The Introduction part has been modified according to the reviewer’s suggestions.

References have been added in the line 67-84.

The line 90-91 has been omitted from the manuscript to avoid any confusion to the readers.

  1. Part 3. Synthesis of plasmonic metal-semiconductor photocatalyst. There is just a short explanation of how the different parameters of synthesized catalysts are affecting the LSPR effect. Then the several methods of the metal-based heterostructure semiconductors just listed, even without the references. And finally just enumeration of the five works where some systems investigated is done. From my opinion this part should be expanded.

Reply: Authors want to thank the reviewer for the comment. The Synthesis part has been modified according to the reviewer’s suggestions.

  1. In averege – from my opinion each numbered section should end with some kind of subconclusion.

Reply: Authors want to thank the reviewer for the comment. Each section has been modified according to the reviewer’s suggestions.

Reviewer 2 Report

The authors tried to summarize the photocatalytic hydrolysis and degradation of toxic dyes by using plasmonic metal-semiconductor heterostructures. The review is timely and good for the researcher. However, some clarifications are needed before it is published.

  1. Typo and grammatical errors should be corrected throughout the manuscript.
  2. A clear gap should be indicated in the introduction.
  3. The abstract is not explanatory for the review. Modification is needed.
  4. Paragraph 1 and 4in the introduction section is not the way of scientific writing. There were no sufficient references. The same is true throughout the manuscript.
  5. The authors should also explain why they want to review on this topic? Why not additional applications?
  6. What makes their review different from other related reviews?
  7. More recent papers with their application on metal and semiconductor-related materials for hydrogen production and dye degradation should be included with their mechanism.
  8. Overall, the review is not written very well. It needs major modification in terms of incorporating the current progress in this research area.

Author Response

We want to thank all reviewers for their valuable comments and suggestions. We have revised the manuscript according to the reviewer’s suggestions. Text and references has been thoroughly revised in the manuscript. Other comments have also been accepted and revised in the manuscript.

Reviewer 2

  1. Typo and grammatical errors should be corrected throughout the manuscript.

Reply: Authors want to thank the reviewer for the comment. All the typos have been corrected in the manuscript.

  1. A clear gap should be indicated in the introduction.

Reply: Authors want to thank the reviewer for the suggestion. The introduction part has been modified.

  1. The abstract is not explanatory for the review. Modification is needed.

Reply: Authors want to thank the reviewer for the suggestion. The abstract has been modified.

  1. Paragraph 1 and 4in the introduction section is not the way of scientific writing. There were no sufficient references. The same is true throughout the manuscript.

Reply: Authors want to thank the reviewer for the suggestion. References have been added in the suggested paragraphs.

  1. The authors should also explain why they want to review on this topic? Why not additional applications?

Reply: Authors want to thank the reviewer for the comment. Hydrogen is a very efficient fuel and is an alternative to petrochemical resources. Therefore world-wide research is going on the production of hydrogen. The photocatalytic hydrolysis of hydrogen storage compounds, in the presence of photocatalysts is a cost-effective and eco-friendly technique to produce hydrogen. Similarly the same photocatalysts are highly efficient in adsorbing toxic dye molecules. The plasmonic property of the metal-semiconductor photocatalyst is a global topic and is very efficient in converting solar energy into triggering above chemical reactions. Addition of all the other applications of plasmonic metal-semiconductor hybrid would have made the review work too lengthy.

  1. What makes their review different from other related reviews?
  2.  

Reply: This review work describes the recent advancement made in the field of photocatalysis using metal-semiconductor hybrid, with emphasis on the significant features of plasmonic metal-semiconductor nanostructures (e.g., size, shape and composition) that influence the optical properties of the photocatalyst. The four essential mechanism of catalytic reaction due to plasmonic metal-semiconductor photocatalyst are also thoroughly explained.  

  1. More recent papers with their application on metal and semiconductor-related materials for hydrogen production and dye degradation should be included with their mechanism.

Reply: Recent papers in the section of hydrogen production and dye degradation have been added.

Round 2

Reviewer 1 Report

The revision submitted by the Authors meets all the points highlighted. Thus I suggest therefore to publish the work as it is, with no further actions from the Authors.

Reviewer 2 Report

The authors now tried to address all the comments. I recommend the manuscript be accepted in its current form.